# Innovation-Driven Policies, Corporate Governance Structure and Total Factor Productivity in Chinese Sports Sector: Evidence from Listed Sports Firms

**Ziyu Guo** 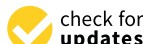**, Gang Chen * and Yang Ding**

Research Center for Sports Strategy and Policy, Wuhan Sports University, Wuhan 430070, China; gzyl009@126.com (Z.G.)
* Correspondence: cg224@163.com

**Abstract:** The sports industry, an emerging industry with low pollution and low emissions, plays an important role in the sustainable development of human society. Using 489 observations from a panel of 128 sports firms listed on the New Third Board in China from 2015 to 2020, this study investigated the effects of three different innovation-driven policies on the total factor productivity of sports firms and the moderating role of governance structure on this relationship. The results showed that high-tech enterprise tax relief was an important policy tool to promote the total factor productivity of sports enterprises, but the direct effects of government subsidies and pre-tax deduction of R&D expenses were not significant. In addition, governance structure had a positive moderating effect on the relationship between innovation-driven policies and the total factor productivity of sports firms. The positive effect of the pre-tax deduction of R&D expenses policy was more significant for sports firms with larger and more independent boards of directors. This study provides new insight into innovation policy development for the sports industry by showing that corporate governance has a significant impact on the effectiveness of innovation-driven policies. Furthermore, the findings provide practical guidance for both managers and government–industry policymakers in the sports industry.

**Keywords:** total factor productivity; innovation-driven policy; governance structure; high-tech sports companies; policy effectiveness

## 1. Introduction

Previous research has documented the social, psychological, economic, and environmental benefits of organized sports activities and sports events [1]. Compared to industries such as manufacturing, mining, and construction, the sports industry is pollution-free and conducive to the sustainable development of mankind. In recent years, the sports industry has become one of the largest and most dynamic pillar industries in some Western countries. However, such strong sports enterprises with independent innovation capability and international competitiveness were lacking in China before 2014, while there was also the problem of inefficient use of resources [2]. This requires promoting continuous innovation in all aspects of production, marketing, and sales for sports companies, thus increasing their total factor productivity (TFP).

TFP is a comprehensive measure of several factors that affect output growth in addition to material inputs such as capital and labor in the production process of an enterprise [3]. Since the size of an enterprise is relatively fixed, TFP can also be seen as the improvement of enterprise efficiency brought about by technological innovation to a certain extent [4]. In order to develop the nascent sports industry, the central government of China released Document No. 46 titled "To Accelerate the Development of Sports Industry and Promote Sports Consumption" in 2014, in which the sports industry was included for the first time in the support scope of innovation-driven policies to improve its total factor productivity. As

policies introduced to match the implementation of China's innovation strategy, innovation-driven policies are designed to promote or serve innovation activities and improve industry-wide innovation capabilities [5]. Under this support, the added value of the Chinese sports industry and its proportion of the GDP have begun to rapidly increase in recent years, becoming an emerging industry (Figure 1).

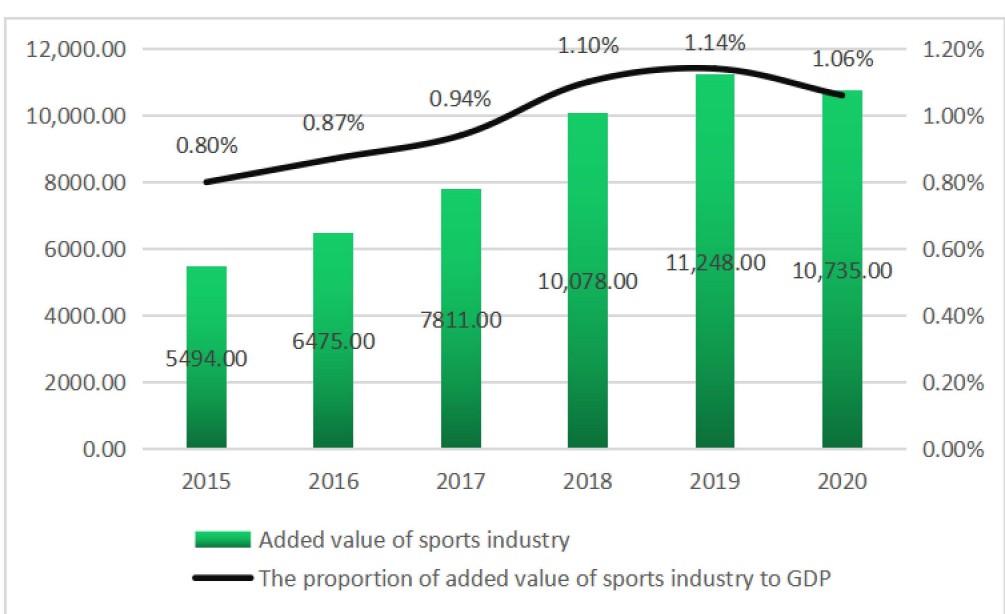

**Figure 1.** Added value of the Chinese sports industry and its share of the GDP from 2015 to 2020.

Generally, innovation-driven policies can not only directly compensate for financial gaps in the production, operation, and innovation activities of enterprises but can also release an economic signal to venture capital companies, financial institutions, etc. [6]. Therefore, they can effectively improve total factor productivity by easing financing constraints and stimulating R&D investment, thus enabling enterprises to develop through technological innovation rather than factor input [7]. However, opponents believe that due to the information asymmetry between the government and the market, the implementation of innovation-driven policies often suffers from policy misallocation [8], which reduces the R&D performance of enterprises and cannot really improve corporate performance. In addition, innovation-driven policies are only external incentives for R&D investment. Some scholars point out that the characteristics of the corporate governance structure also have an impact on enterprise risk-taking and R&D decisions [9,10], which is particularly obvious in China's sports industry dominated by medium- and minor-sized enterprises [11]. As a result, the effectiveness of innovation-driven policies to promote the total factor productivity of sports firms with different governance structures has become a pressing issue that requires further research.

Specifically, we identified four research gaps that could be filled to gain a better understanding of the development and implementation of innovation-driven policies. Firstly, the effect of innovation-driven policies on total factor productivity is still controversial. Secondly, the relationship between governance structure and sports firms' total factor productivity is underexplored. Thirdly, it remains unclear how governance structure moderates the effects of industry policies. Lastly, previous studies have focused on innovation-driven policies for traditional industries while lacking attention to emerging industries such as the sports industry.

Accordingly, using 489 observations from a panel of 128 sports firms listed on the New Third Board in China from 2015 to 2020, this study aimed to fill the knowledge gaps by addressing the following three questions:

1. Do innovation-driven policies promote or inhibit improvement of the total factor productivity of Chinese sports enterprises?
2. What is the relationship between the governance structure of sports enterprises and their total factor productivity?
3. How does governance structure influence the process of innovation-driven policies affecting the total factor productivity of sports enterprises?

The results of this study may have important impacts on the management of R&D activities within sports companies and the implementation of governmental innovation-driven policies. These revelations include: (1) evaluating the effectiveness of innovation-driven policies to provide a reference for future policy formulation; (2) adopting an appropriate governance structure to improve the innovation efficiency of sports enterprises and the use of policies; (3) identifying the moderating effect of governance to facilitate the precise supply of innovation-driven policies; and (4) studying the law of technological innovation in China's sports industry. The research conclusions are of reference value to the innovation management of sports enterprises in other countries or regions and new enterprises or emerging industries.

In the sections that follow, we first develop a theoretical framework regarding the relationship between innovation-driven policies, corporate governance structure, and total factor productivity. Then, we test the hypotheses through a series of regression models. Finally, we conclude with a thorough discussion of the implications of this study as they relate to both theory and practice.

## 2. Theoretical Framework and Hypothesis Development

In recent years, total factor productivity, an important indicator for evaluating efficiency, has become a criterion for judging whether a growth model has been transformed into a sustainable growth model [12]. However, empirical studies show that the sports, culture, and entertainment industries not only have low financing efficiency, but their total factor productivity is also in the worst position [13]. This requires not only effective public expenditure policies from the government [14] but also internal quality control by the sports companies themselves [15]. Accordingly, this study empirically examined the impact of innovation-driven policies on the total factor productivity of sports firms and the moderating role of governance structure.

### 2.1. Innovation-Driven Policies Impacting Total Factor Productivity

The measurement of TFP cannot be separated from related financial support and guarantees from the government [16]. Government support for firms often aims to improve their economic performance and productivity through a combination of direct and indirect funding, tax incentives, special loans, and other similar policies [17].

2.1.1. Government Subsidies and Total Factor Productivity

Government subsidies are one source of funding to support the innovation efforts of firms [18]. On the one hand, as a direct capital injection, government subsidies provide enterprises with the means to rapidly accumulate innovative resources and reduce R&D costs so they can alleviate the financial pressures on enterprise R&D, thus significantly raising the level of TFP by improving efficiency through technological improvement [19]. On the other hand, subsidies also reflect a government's positive attitude towards recipient companies [20]; therefore, they can produce a kind of "certification effect" [21] that helps firms to expand their external collaborations and address problems associated with R&D underinvestment, which enables additional knowledge and support acquisition for further innovation.

2.1.2. Pre-Tax Deduction of R&D Expenses and Total Factor Productivity

Firms' innovation activities are sensitive to specific R&D-related taxation changes. Pre-tax deduction of R&D expenses can not only change the relative cost between R&D

investment and factor input, thus reducing the capital costs of enterprise R&D activities, but it also helps firms to determine the direction of R&D expenditure [22], making them more inclined to conduct technological innovation rather than invest in factor input and improving their total factor productivity [23]. However, as a typical selective tax incentive, the implementation of an R&D expense deduction policy may easily lead to adverse selection behaviors such as "rent-seeking," R&D manipulation, and strategic innovation by some enterprises, which may reduce the effect of the policy [24].

2.1.3. High-Tech Enterprise Tax Relief and Total Factor Productivity

High-tech enterprise tax relief is another important innovation-driven policy through which the government can encourage innovation and R&D [25]. In China, resident enterprises that have been registered for more than one year can apply to the administrative departments of science and technology, finance, and taxation for recognition as high-tech enterprises and thus enjoy tax benefits. As a kind of tax-based preferential method, high-tech enterprise tax relief can bring direct income effects to enterprises, i.e., it provides financial support for enterprises to help expand reproduction and technological innovation through channels such as optimizing resource allocation, easing financing constraints, and improving the net profit after tax, thus promoting the improvement of total factor productivity [26].

Based on the above analysis, it can be seen that innovation-driven policies can stimulate firms to innovate, which is one of the important ways to improve the total factor productivity of firms. Currently, the development of China's sports industry is still in the preliminary phase, and many new enterprises have launched in the past five years. Such companies often have greater financial pressure to engage in innovation activities with high investment and long lead times, which requires external incentives from the government. Therefore, whether innovation-driven policies are effective for the Chinese sports industry needs be empirically examined. This led to our first research question and the following hypotheses:

RQ1: How do innovation-driven policies affect the total factor productivity of Chinese sports firms?

**Hypothesis 1 (H1).** *Government subsidies significantly and positively affect the TFP of sports firms.*

**Hypothesis 2 (H2).** *Pre-tax deduction of R&D expenses significantly and positively affects the TFP of sports firms.*

**Hypothesis 3 (H3).** *High-tech enterprise tax relief significantly and positively affects the TFP of sports firms.*

*2.2. Impact of Governance Structure on Total Factor Productivity*

Corporate governance is the structure by which the interests of owners are aligned to those of managers. It often provides monitoring arrangements for owners or an incentive system to persuade managers to take risks and increase the firm's value [10,27]. For the Chinese sports industry, governance structure is an important factor impacting the technological innovation of sports enterprises; it includes the ownership structure, board size, and ratio of independent directors [11].

2.2.1. Equity Concentration and Total Factor Productivity

Equity concentration refers to the quantification of centralized or decentralized ownership due to different shareholding ratios among all shareholders. Its level determines the weighing and supervision of strong shareholders on entrepreneurial business behavior decisions, especially on R&D behavior [28]. For corporations, keeping an equity concentration can increase the major shareholders' control over the company and help to alleviate the agency problem [29]. As the largest shareholder's shareholding ratio rises,

strong shareholders will more actively join the process of company strategic decision-making, actively establish an effective and independent governance system, and effectively supervise management business operations [30]; the regulatory benefits far exceed the corresponding regulatory costs as the company's performance improves [31]. When the largest shareholder has complete control, the higher its shareholding ratio, the better the company's performance [32].

### 2.2.2. Board Size and Total Factor Productivity

Generally, increasing the number and diversity of the board of directors can expand the company's social network, build a broad resource platform, help companies to achieve complementary advantages, and improve corporate performance [33]. However, as board size expands, internal disputes and differences of opinion increase and poor communication and decision-making issues can reduce the effectiveness of large boards [34]. Additionally, higher communication and coordination costs will make it more difficult for firms to hold board meetings or reach consensus, and therefore tend to reduce corporate risk taking [35]. This is why some scholars suggest that firms strengthen corporate governance by reducing the size of the board of directors [36].

### 2.2.3. Board Independence and Total Factor Productivity

Board independence can be measured by the percentage of independent directors on the board [37]. As experts and supervisors on the board of directors, independent directors can influence corporate performance in three ways. Firstly, independent directors can take advantage of information to avoid managerial misconduct and emptying by major shareholders through effective supervision [38]. Secondly, independent directors are able to provide professional advice as experts on enterprise operation [39]. Lastly, independent directors with special background characteristics can help companies to coordinate relationships and acquire information, thus improving the ability to obtain resources and the stability of the company. All of these are important to enhance the total factor productivity of the company.

Based on the above analyses, corporate governance structure affects the TFP of companies. In China, sporting equipment manufacturing is the largest sports-related industry in terms of production value, and the vast majority of these firms are private enterprises, most of which adopt a family management model. However, compared with American family businesses, which have high surplus quality, low cost of debt, and enjoy a family control premium, Chinese family businesses are not as developed and in many ways are not regulated [40]. Thus, Chinese sports firms are more likely to need larger board sizes and more independent directors to strengthen management. This led to the second research question and Hypotheses 4–6:

RQ2: How does corporate governance structure affect the total factor productivity of Chinese sports firms?

**Hypothesis 4 (H4).** *Equity concentration significantly and positively affects the TFP of sports firms.*

**Hypothesis 5 (H5).** *Board size significantly and positively affects the TFP of sports firms.*

**Hypothesis 6 (H6).** *Board independence significantly and positively affects the TFP of sports firms.*

### 2.3. Moderating Effects of Governance Structure on the Relationship between Innovation-Driven Policies and Total Factor Productivity

The purpose of the government's implementation of innovation-driven policies is to activate the innovation vitality of companies by reducing the costs of innovation and improving innovation income. However, companies of similar size, influence, and support may have very different innovation performances under the same external conditions. This shows that the operation, management, and innovation activities of companies are

conducted under their established governance structures, and the governance structure is an important factor that determines the efficiency of innovation-driven policies and the innovation performance of sports companies [41].

Unlike companies with strong governance structures, companies with weak governance structures tend to have more serious agency problems and more moral hazard problems [42]. That is, when agents face limited investment opportunities, they may adopt "short-sighted" behavior, becoming more inclined to reduce R&D investment, hold more cash, or use government subsidies for non-R&D purposes in order to enhance the corporation's influence and reduce market constraints [43]. However, good company governance can help to fundamentally optimize the internal management, resource allocation, and decision-making mechanisms [44] so that companies can make full use of the financial support provided by innovation-driven policies and invest in technological innovation and product R&D rather than chasing short-term goals, thus strengthening fiscal and taxation support for company innovation. For example, concentration of ownership can reduce the degree of information asymmetry and improve decision-making governance [45], while outside directors can independently encourage innovation through compensation incentives for senior managers [46].

Based on the above analysis, governance structure likely plays an important role in the relationship between innovation-driven policies and company efficiency improvement. For this reason, this study introduced an interaction term, the product of innovation-driven policy data and corporate governance structure data, with reference to the study by Chen et al. [11] on the internal endowment of sports firms. This led to RQ3 and the following hypotheses:

RQ3. How does governance structure influence the process of innovation-driven policies affecting the total factor productivity of sports enterprises?

**Hypothesis 7 (H7).** Governance structure positively moderates the association between innovation-driven policies and the TFP of sports firms.

Based on the above literature, the theoretical framework of this study is presented in Figure 2.

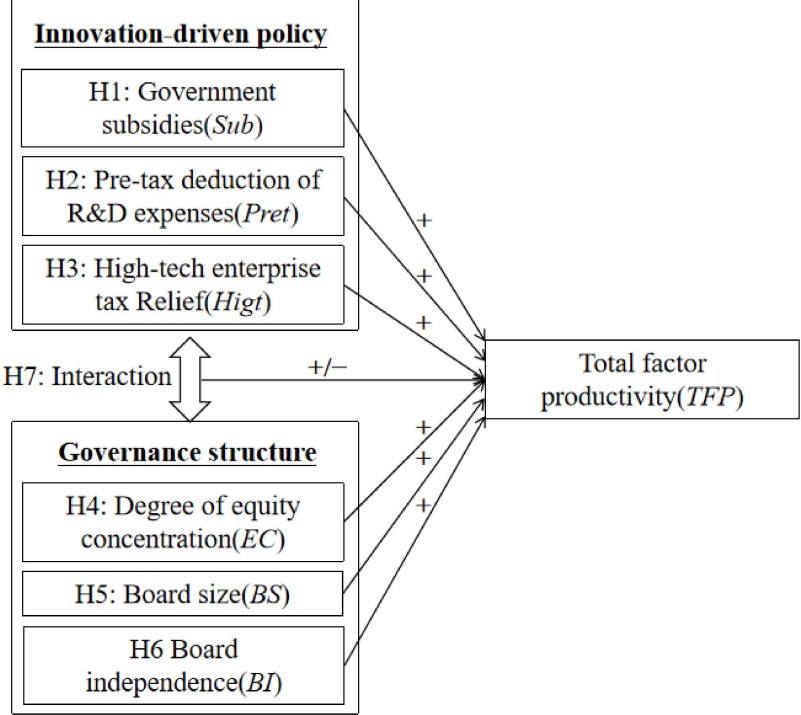

**Figure 2.** Theoretical Framework.

## 3. Method

### 3.1. Research Context and Data

The New Third Board, a stock transfer trading venue established in 2006, targets small- and medium-sized enterprises in China. Compared with the main boards, the SME and GEM boards, the listing requirements of the New Third Board are substantially relaxed in many aspects, such as qualifications, financial index, assets, share capital, corporate governance, etc. Therefore, it is more inclusive for innovative and growing but not yet profitable enterprises.

Since Document No. 46 was released in 2014, the sports industry, as a rapidly developing sunrise industry in China, has gained the favor of investors. As a result, more and more sports firms have been listed on the New Third Board and have become an important driving force in the innovative development of China's sports industry. Therefore, this study used related data from sports firms listed on the New Third Board as a research sample. To exclude errors in the study results caused by the macro policy environment and the effects of the COVID-19 pandemic, the timespan of this study was 2015–2020.

In order to test the relationship between innovation-driven policies, corporate governance structure, and the total factor productivity of sports firms, we created a panel of sports firms listed on the New Third Board for which the main business involved sports manufacturing and services. In total, we found that there were 128 sports firms listed on the New Third Board from 2015 to 2020. In selecting the data, several firms had missing values for certain indicators for a number of reasons (e.g., operating anomalies during the study period; inability to properly disclose annual reports); thus, the final sample was an unbalanced panel including 128 companies, with 489 observations. Data for each company were obtained from the annual reports of listed companies, which were obtained manually. Data processing and analysis were performed in Stata/SE 15.

### 3.2. Research Variables

Table 1 summarizes the variables included in the current study.

**Table 1.** Indicators of innovation-driven policies and corporate governance structures.

| Variables (Abbreviation) | Measurement | Reference |
|---|---|---|
| Total factor productivity (TFP) | Total factor productivity is evaluated based on $BC^2$ mode | Zhu et al., 2014 [47] Chen, 2014 [48] |
| Government subsidies (Sub) | Total government subsidy to support firms | Zhang and Guan, 2018 [18] |
| Pre-tax deduction of R&D expenses (Pret) | Pre-tax deduction of R&D expenses | Chen et al., 2020 [5] |
| High-tech enterprise tax relief (Higt) | Income tax relief amount of high-tech enterprise | Ding and Chen, 2022 [49] |
| Equity concentration (EC) | Shareholding of the top five largest shareholders | |
| Board size (BS) | Number of members on board of directors | Chen et al., 2019 [11] |
| Board independence (BI) | Percentage of independent directors | |
| Scale of asset (ASS) | Total assets | |
| Asset–liability ratio (LEV) | Debt-to-asset ratio | Chen et al., 2021 [50] |
| Intensity of R&D staff (RDP) | R&D staff ratio | |

### 3.2.1. Dependent Variable

The dependent variable was total factor productivity ("TFP"), and its value was evaluated based on the $BC^2$ model. Compared to the traditional $C^2R$ model, the $BC^2$ model takes into account the case of variable returns to scale, and the calculation of technical efficiency is not affected by economies of scale. In terms of specific variable selection, with reference to the existing literature, this paper used total assets to measure enterprise capital inputs, enterprise employees to measure enterprise labor inputs, and operating income as the output variable.

### 3.2.2. Independent Variables

The independent variables were the three types of innovation-driven policies, including government subsides ("*Sub*"), pre-tax deduction of R&D expenses ("*Pret*"), and high-tech enterprise tax relief ("*Higt*"). Specifically, "*Sub*" and "*Pret*" were measured by the ex ante financial subsidies and ex post-tax deduction of R&D expenses, which are offered to support sports firms to engage in innovation activities. "*Higt*" is a binary indicator that reflects whether the firm is a high-tech enterprise. Enterprises recognized as high-tech enterprises can enjoy a 15% income tax deduction, which is 10% lower than the normal tax rate. Thus, "*Higt*" can be measured as 10% of a high-tech firm's net profit, while "*Higt*" is equal to 0 if a firm is in the red for the year.

### 3.2.3. Moderating Variables

The moderating variables were the governance structure of sports firms, which was measured by equity concentration ("*EC*"), board size ("*BS*"), and board independence ("*BI*"). Specially, "*EC*" was equal to the shareholding of the top five largest shareholders, "*BS*" was equal to the number of members on the board of directors, and "*BI*" was measured by a percentage of the independent directors.

### 3.2.4. Control Variables

In addition, in line with general practice, this study controlled for various factors that may influence the extent to which two innovation-driven policies affect the market value of sports firms, including scale of asset ("*ASS*"), asset–liability ratio ("*LEV*"), and intensity of R&D staff ("*RDP*").

### *3.3. Empirical Models*

RQ1 explored the impact of innovation-driven policies on the total factor productivity of sports firms, which in turn revealed the effect of external factors on total factor productivity. To examine Hypotheses H1–H3, the following regression model was developed based on the above literature reviews:

$$TFP = \delta_1^1 Sub + \delta_2^1 Pret + \delta_3^1 Higt + \delta_4^1 ASS + \delta_5^1 LEV + \delta_6^1 RDP + \vartheta_1 \tag{1}$$

In Equation (1), *Sub* is the government subsidies obtained by the enterprise; *Pret* is the total amount of pre-tax deduction of R&D expenses enjoyed by the enterprise; *Higt* is the tax relief obtained by the high-tech sports enterprise; *ASS, LEV,* and *RDP* are control variables representing the scale of the enterprise's assets and solvency and the proportion of enterprise R&D personnel, respectively.

$$TFP = \delta_1^2 EC + \delta_2^2 BS + \delta_3^2 BI + \delta_4^2 ASS + \delta_5^2 LEV + \delta_6^2 RDP + \vartheta_2 \tag{2}$$

RQ2 explored the impact of governance structure on the total factor productivity of sports firms, which in turn revealed the effect of internal factors on total factor productivity. To address RQ2, we developed the following regression model:

$$TFP = \delta_1^3 Sub + \delta_2^3 Pret + \delta_3^3 Higt + \delta_4^3 EC + \delta_5^3 BS + \delta_6^3 BI + \delta_7^3 ASS + \delta_8^3 LEV + \delta_9^3 RDP + \vartheta_3 \tag{3}$$

Equations (1) and (2) show that innovation-driven policies and corporate governance structure directly affect total factor productivity. Based on these equations, Equation (3) shows how innovation-driven policies and corporate governance structure affect total factor productivity together, demonstrating the compound effect of both internal and external factors on the decision of total factor productivity. The exploration of RQ1 and RQ2 led to the development of the following regression model.

$$TFP = [\delta_1^4][Policies] * [Governance] + [Control] + \vartheta_4 \tag{4}$$

In Equation (4), [Policies] and [Governance] reflect the innovation-driven policies ("*Sub*", "*Pret*", "*Higt*") and governance structure ("*EC*", "*BS*", "*BI*") that are significantly related to total factor productivity based on the results of the primary regressions; likewise, [Control] is similar to Equations (1)–(3) ("*ASS*", "*LEV*", "*RDP*"). To exclude the bias of the estimation results due to the heteroskedasticity problem, all the above continuous type variables were treated as logarithmic.

## 4. Results

### 4.1. Descriptive Statistics

Table 2 shows the current sports firms listed on the New Third Board and further describes the research context. According to Table 2, the most support received by sports firms came from government subsidies, whereas the least support received by sports firms was from pre-tax deduction of R&D expenses. Further analysis found that 18.84% of high-tech sports firms did not qualify for pre-tax deduction of R&D expenses. The mean value of EC was large, the mean value of BI was small, and they both had small variances with BS. This was likely because most sports firms (78.9%) only had five directors, and very few firms (4.09%) had an independent director. This meant that most sports firms' governance structures were very simple and lacked openness and variety. It is worth mentioning that TFP was significantly and positively correlated with all three types of innovation-driven policies, which was consistent with our hypothesis and suggested that these policies may be beneficial in supporting productive innovation outcomes and higher total factor productivity for these firms.

**Table 2.** Results of descriptive statistics.

| | Mean | Std.Dev. | Min | Max | 1 | 2 | 3 | 4 | 5 | 6 | 7 | 8 | 9 | 10 |
|---|---|---|---|---|---|---|---|---|---|---|---|---|---|---|
| 1. TFP | 0.149 | 0.165 | 0.000 | 1.000 | 1.000 | | | | | | | | | |
| 2. Sub | 10.879 | 5.536 | 0.000 | 17.836 | 0.091 | 1.000 | | | | | | | | |
| 3. Pret | 3.622 | 5.879 | 0.000 | 16.516 | 0.134 | 0.269 | 1.000 | | | | | | | |
| 4. Higt | 5.323 | 6.835 | 0.000 | 18.477 | 0.278 | 0.230 | 0.302 | 1.000 | | | | | | |
| 5. EC | 0.900 | 0.114 | 0.500 | 1.000 | −0.231 | −0.057 | −0.012 | −0.070 | 1.000 | | | | | |
| 6. BS | 1.688 | 0.189 | 0.693 | 2.565 | 0.167 | 0.133 | −0.042 | 0.161 | −0.334 | 1.000 | | | | |
| 7. BI | 0.015 | 0.074 | 0.000 | 0.600 | 0.197 | 0.110 | 0.053 | 0.083 | −0.252 | 0.316 | 1.000 | | | |
| 8. ASS | 18.238 | 1.416 | 13.561 | 22.454 | −0.179 | 0.009 | 0.026 | −0.152 | 0.118 | −0.077 | −0.056 | 1.000 | | |
| 9. LEV | 1.239 | 6.502 | 0.000 | 87.090 | −0.042 | −0.071 | −0.077 | −0.103 | 0.022 | −0.107 | −0.027 | −0.032 | 1.000 | |
| 10. RDP | 0.196 | 0.206 | 0.000 | 0.953 | 0.033 | 0.045 | 0.034 | 0.236 | −0.041 | 0.088 | −0.053 | −0.055 | −0.086 | 1.000 |

### 4.2. Effects of Innovation-Driven Policies on Total Factor Productivity of Sports Firms

RQ1 explored which innovation-driven policies were associated with total factor productivity. As shown in Table 3, the first and third columns present the results of models 1 and 3, which tested H1, H2, and H3 (i.e., innovation-driven policies significantly and positively affect the total factor productivity of sports firms).

The results of model 1 indicated that innovation-driven policies directly explained 10.22% of the variance in firms' total factor productivity (F (6482) = 9.15, $p < 0.01$). Meanwhile, the results of model 3 indicated that innovation-driven policies and corporate governance structure together explained 15.65% of the variance in firms' total factor productivity (F (9479) = 9.88, $p < 0.01$). Both models 1 and 3 found that *Higt* was positively related to *TFP*, but *Sub* and *Pret* were not significantly related to *TFP*. Thus, H3 was supported, whereas H1 and H2 were not supported.

**Table 3.** The effects of innovation-driven policies on total factor productivity of sports firms.

| | Model 1 TFP | Model 2 TFP | Model 3 TFP |
|---|---|---|---|
| Sub | 0.001 (0.001) | | 0.000 (0.001) |
| Pret | 0.002 (0.001) | | 0.002 (0.001) |
| Higt | 0.006 *** (0.001) | | 0.005 ***(0.001) |
| EC | | −0.233 *** (0.068) | −0.233 *** (0.066) |
| BS | | 0.049 (0.042) | 0.033 (0.041) |
| BI | | 0.291 *** (0.103) | 0.255 ** (0.100) |
| ASS | −0.017 *** (0.005) | −0.017 *** (0.005) | −0.014 *** (0.005) |
| LEV | 0.000 (0.001) | −0.001 (0.001) | 0.000 (0.001) |
| RDP | −0.029 (0.036) | 0.014 (0.035) | −0.026 (0.035) |
| _cons | 0.424 *** (0.095) | 0.587 *** (0.139) | 0.529 *** (0.135) |
| F-value | 9.15 *** | 8.98 *** | 9.88 *** |
| $R^2$ | 0.1022 | 0.1006 | 0.1565 |

Note: Dependent variable is TFP; the standard error is noted in parenthesis (); ** $p < 0.05$, *** $p < 0.1$.

### 4.3. Effect of Governance Structure on Total Factor Productivity of Sports Firms

RQ2 explored which governance structure was associated with total factor productivity. As shown in Table 3, the second and third columns present the results of models 2 and 3, which tested H4, H5, and H6 (i.e., corporate governance structure significantly and positively affects the total factor productivity of sports firms).

The results of model 2 indicated that corporate governance structure directly explained 10.06% of the variance in firms' total factor productivity (F (6482) = 8.98, $p < 0.01$). Meanwhile, the results of model 3 indicated that innovation-driven policies and corporate governance structure together explained 15.65% of the variance in firms' total factor productivity (F (9479) = 9.88, $p < 0.01$). Both models 2 and 3 found that *BI* was positively related to *TFP*, *EC* was negatively related to *TFP*, and *BS* was not significantly related to *TFP*. Thus, H6 was supported, H4 and H5 were not supported, but the inverse proposition of H4 was supported.

### 4.4. Effect of the Interaction between Innovation-Driven Policies and Corporate Governance Structure on Total Factor Productivity of Sports Firms

RQ3 concerned the moderating effects of governance structure on the relationship between innovation-driven policies and total factor productivity. As shown in Table 4, models 4, 5, and 6 regressed the firms' total factor productivity on three different interaction terms between innovation-driven policies ("*Sub*," "*Pret*," "*Higt*") and corporate governance structure ("*EC*", "*BS*," "*BI*").

Model 4 (F (6482) = 7.03, $p < 0.01$) showed that the interaction between "*EC*" and "*Higt*" was significantly and positively related to *TFP*. Model 5 (F (6482) = 11.03, $p < 0.01$) showed that the interaction between "*BS*," "*Pret*," and "*Higt*" was significantly and positively related to *TFP*. Model 6 (F (6482) = 11.88, $p < 0.01$) showed that the interaction between "*BI*" and "*Pret*" was significantly and positively related to *TFP*. Thus, H7 was supported.

**Table 4.** The effect of the interaction between innovation-driven policies and corporate governance structure on total factor productivity of sports firms.

| | Model 4 TFP | Model 5 TFP | Model 6 TFP |
|---|---|---|---|
| EC-Sub | −0.001 (0.001) | | |
| EC-Pret | 0.001 (0.001) | | |
| EC-Higt | 0.006 *** (0.001) | | |
| BS-Sub | | 0.001 (0.001) | |
| BS-Pret | | 0.001 * (0.001) | |
| BS-Higt | | 0.003 *** (0.001) | |
| BI-Sub | | | 0.002 (0.010) |
| BI-Pret | | | 0.077 *** (0.016) |
| BI-Higt | | | 0.005 (0.015) |
| ASS | −0.018 *** (0.005) | −0.016 *** (0.005) | −0.019 *** (0.005) |
| LEV | −0.001 (0.001) | 0.000 (0.001) | −0.001 (0.001) |
| RDP | −0.022 (0.036) | −0.036 (0.035) | 0.002 (0.034) |
| _cons | 0.460 *** (0.096) | 0.406 *** (0.095) | 0.481 *** (0.092) |
| F-value | 7.03 *** | 11.03 *** | 11.88 *** |
| $R^2$ | 0.080 | 0.121 | 0.129 |

Note: Dependent variable is TFP; the standard error is noted in parenthesis (); * $p < 0.01$, *** $p < 0.1$.

### 4.5. Robustness Tests

In order to test the reliability of the empirical results of this study, the following robustness tests were used: (1) pure technical efficiency was selected as the dependent variable, which was evaluated based on the BC$^2$ model; (2) the independent variable was set as a dummy variable, with sports firms enjoying policy support set as "1", otherwise it was set as "0"; (3) high-tech enterprises were retained as a sub-sample; (4) all variables were winsorized at the 1% quantile. The results of these robustness tests are reported in Table 5. It can be seen that the regression results did not change substantially, indicating that the findings of this study had strong robustness.

**Table 5.** Results of robustness tests.

| | Model 7 TFP | Model 8 TFP | Model 9 TFP | Model 10 TFP |
|---|---|---|---|---|
| Sub | −0.000 (0.002) | −0.016 (0.022) | −0.003 (0.004) | −0.000 (0.001) |
| Pret | 0.002 (0.002) | 0.018 (0.020) | 0.002 (0.002) | 0.002 (0.001) |
| Higt | 0.005 *** (0.001) | 0.050 *** (0.019) | 0.069 *** (0.011) | 0.005 *** (0.001) |
| EC | −0.292 *** (0.776) | −0.299 *** (0.078) | −0.097 (0.160) | −0.243 *** (0.068) |

**Table 5.** *Cont.*

| | Model 7 TFP | Model 8 TFP | Model 9 TFP | Model 10 TFP |
|---|---|---|---|---|
| BS | −0.057 | −0.047 | −0.081 | 0.014 |
| | (0.049) | (0.049) | (0.082) | (0.045) |
| BI | 0.285 ** | 0.296 ** | 0.257 | 0.292 *** |
| | (0.118) | (0.119) | (0.184) | (0.111) |
| ASS | −0.030 *** | −0.031 *** | −0.031 *** | −0.014 *** |
| | (0.006) | (0.006) | (0.010) | (0.005) |
| LEV | 0.001 | 0.001 | 0.121 * | −0.001 |
| | (0.001) | (0.001) | (0.069) | (0.002) |
| RDP | −0.009 | 0.001 | 0.098 | −0.024 |
| | (0.041) | (0.041) | (0.066) | (0.035) |
| _cons | 1.047 *** | 1.061 *** | −0.032 | 0.569 *** |
| | (0.159) | (0.160) | (0.321) | (0.143) |
| F-value | 9.31 *** | 8.46 *** | 9.54 *** | 9.77 *** |
| $R^2$ | 0.1488 | 0.1372 | 0.3266 | 0.1551 |

Note: Dependent variable is TFP; the standard error is noted in parenthesis (); * $p < 0.01$, ** $p < 0.05$, *** $p < 0.1$.

## 5. Discussion, Conclusions, and Suggestions

### 5.1. Discussion and Conclusions

This study empirically examined how innovation-driven policies impact the total factor productivity of sports firms and the moderating role of governance structure. The results indicated that innovation-driven policies and corporate governance structure both affect the total factor productivity of sports firms, and corporate governance structure can positively moderate the effects of innovation-driven policies.

To begin, it was apparent that government subsidies did not have a significant relationship with the total factor productivity of sports firms. This finding was different from Carboni [7], who found that public grants had a positive effect both on firm investment and R&D. Our findings suggested that government subsidies did not improve the total factor productivity of sports firms. This phenomenon could be explained by the following reasons: For one thing, due to the complexity of technological innovation, there is general information asymmetry between the government and the market. Chinese enterprises often provide packaged information for subsidy applications through hired university researchers, exaggerate the value of projects, etc. [51]. Enterprises that receive subsidies in this manner are frequently ill-suited for frontier technology innovation projects, which wastes subsidy funds and also reduces the efficiency of internal resource allocation by enterprises. For another, according to Lee et al. [20], the effect of government subsidies is closely related to the purpose of subsidized enterprises. When R&D funds come from the government rather than the enterprises themselves, enterprise R&D departments will not be eager to urgently transform R&D results into innovative outputs in order to compensate for their own preliminary investment, which results in the moral hazard of capital abuse. Both of the above-mentioned points will cause the problem of "policy mismatch" and thus reduce the effectiveness of government subsidies.

Continuing, we found that the impact of pre-tax deduction of R&D expenses on total factor productivity was positive but not significant. This finding was similar to Chen and Breedlove [5], who found that pre-tax deduction of R&D expenses did not have a significant effect on the total factor productivity of sports firms. There are three explanations for this phenomenon: First, the support provided to enterprises by the pre-tax deduction of R&D expenses is linked to the R&D expenditure of enterprises. Chinese sports enterprises with small asset size, weak cash flow, and less R&D expenditure receive fewer tax incentives, which are insufficient to alleviate financing constraints in the process of R&D activities [52]. Second, pre-tax deduction of R&D expense is a post-event policy [53], which lacks beforehand and process guidance for enterprises. Thus, enterprises are likely inclined towards projects with fast investment return rather than long-term high-tech

projects, which reduces their innovation performance. Third, pre-tax deduction of R&D expenses is a developing policy in China, and its supporting scope and standards are not fixed. In our sample, 71.98% of the companies were not supported by the policy; the actual situation was far below the expected coverage of the policy.

Nonetheless, high-tech enterprise tax relief was significantly and positively related to the total factor productivity of sports firms. This finding was similar to the conclusion reached by Kasahara et al. [23], who found that special loans and tax credits positively affected firm performance. In China, high-tech sports enterprises can obtain more financial support from the high-tech enterprise tax relief policy. Meanwhile, the government's management of high-tech enterprises is dynamic and high-tech enterprise qualifications are reviewed every three years based on R&D investment and output [5]. For this reason, high-tech enterprises attach great importance to technological innovation. Combined with the fact that the effect of pre-tax deduction of R&D expenses on the total factor productivity of sports firms was not significant, the results indicate that the target audience of sports industry policies has a significant impact on the policies' effects.

Additionally, we found that among corporate governance structures, board size was not significantly related to firm TFP, whereas equity concentration and board independence were significantly related to TFP. This suggests that both shareholders and boards of directors are important factors directly influencing the TFP of Chinese sports firms. Specifically, equity concentration negatively affected the total factor productivity of sports firms, which may be due to the high equity concentration of Chinese sports firms in general and the existence of a large number of family-owned firms, which tend to be more risk averse and more likely to pursue sub-optimal, low-risk investments since family owners invest most of their wealth in business operations [54]. Additionally, board independence positively affected the total factor productivity of firms, which confirmed the view of Cheng et al. [38] that independent directors, as non-controlling shareholders, may improve corporate governance by giving play to their own information strengths and acting as effective supervisors.

In addition, governance structure, especially board characteristics, had a positive moderating effect in the process of innovation-driven policies promoting firm efficiency. It was found that the interaction variables of board size, board independence, and pre-tax deduction of R&D expenses significantly and positively affected the total factor productivity of sports firms. This suggests that the pre-tax deduction of R&D expense policy works for firms with better board governance. In China, sports firms are mostly family-owned and have simple governance structures. An increase in board size and independence means that directors with different backgrounds and experiences are included [55], allowing decision making to be based on future earnings, which can facilitate innovation strategies, expand policy deductions, and ultimately contribute to improved firm performance. It is worth noting that the interaction variables of board independence and high-tech enterprise tax relief did not have a significant relationship with firm total factor productivity, suggesting that independent directors on the boards of Chinese sports firms do not play a role in expanding firm access to resources and information-gathering capabilities. In fact, 60% of the firms in the sample with independent directors did not receive high-tech enterprise recognition, thus reducing the actual supportive effect of the policy.

### 5.2. Suggestions

To improve total factor productivity, Chinese sports enterprises should not only pay attention to technological innovation but also focus on building a governance system to match it. On the one hand, sports firms should appropriately reduce equity concentration, break the family governance mechanism, establish equity checks and balances and corporate agency mechanisms, remove constraints on R&D investment, and enhance management's self-confidence and motivation to engage in technological innovation activities. On the other hand, sports enterprises should continuously optimize the shareholder and board structure, expand the proportion of independent directors with different industry and

technical backgrounds, strengthen the efficiency of internal corporate governance as well as the efficiency of using external policies, and improve corporate innovation performance.

To promote the total factor productivity of sports enterprises, the Chinese government should constantly improve innovation-driven policies. The findings of this paper emphasize the effectiveness of tax incentives in promoting innovation in sports enterprises. Therefore, the Chinese government should further improve the scope of application and preferential treatment of the pre-tax deduction of R&D expenses policy and the recognition of high-tech enterprises so as to boost the enthusiasm and initiative of enterprises for R&D and innovation, thereby comprehensively and sustainably improving their total factor productivity. Meanwhile, in order to optimize the effect of government subsidies on promoting innovation in sports enterprises, the Chinese government should carry out special funding programs to support the development of innovation in the sports industry, strengthen the process management and guidance of technological innovation in supported sports enterprises, and actively implement post-implementation funding programs for successful innovation projects.

Finally, the evidence suggests that the effects of innovation-driven policies vary depending on the governance structure of sports firms. Therefore, to improve the positive effect of innovation-driven policies on the total factor productivity of sports firms, the Chinese government should adopt differentiated innovation-driven policy support for sports enterprises with different resource endowments, focusing on the simultaneous use of different policy tools and making timely dynamic adjustments to optimize the response to changes in the market environment.

## 6. Research Significance and Prospects

This study empirically examined the impact of innovation-driven policies on the total factor productivity of sports firms and the moderating role of governance structure. On the one hand, the findings enrich the theory related to industrial policy, especially highlighting the important influence of governance structure on firm innovation as well as on policy implementation, and the research provides an important complement to the study of the factors influencing policy efficiency. On the other hand, this study examines the effects of different innovation-driven policies and provides a reliable decision reference for government departments to further improve innovation-driven policies and promote innovative development in the sports industry.

Due to the impact of the COVID-19 pandemic on the sports industry, a large number of sports firms closed down, were delisted, and were unable to release their annual reports as usual, thus this study is based on panel data only until 2020 and cannot study the innovative activities of sports firms in the context of the epidemic. However, the impacted sports enterprises are in greater need of financial support from the government. For this reason, the authors will further extend the research schedule in follow-up work. The special characteristics and innovation-driven mechanisms of the sports industry need to be further investigated.

**Author Contributions:** Conceptualization, Z.G. and G.C.; methodology, Z.G.; software, Z.G.; validation, Z.G.; formal analysis, Z.G.; investigation, Z.G.; resources, Z.G. and G.C.; data curation, Z.G.; writing—original draft preparation, Z.G.; writing—review and editing, Z.G., G.C. and Y.D.; visualization, Z.G.; supervision, G.C. and Y.D.; project administration, G.C.; funding acquisition, G.C.All authors have read and agreed to the published version of the manuscript.

**Funding:** This research was funded by the project of the National Social Science Fund of China (No. 21FTYB007).

**Institutional Review Board Statement:** Not applicable.

**Informed Consent Statement:** Not applicable.

**Data Availability Statement:** Data available on request from the authors.

**Acknowledgments:** The authors would like to thank the anonymous reviewers for their valuable comments.

**Conflicts of Interest:** The authors declare no conflict of interest.

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
