# Peer review of "Innovation-Driven Policies, Corporate Governance Structure and Total Factor Productivity in Chinese Sports Sector: Evidence from Listed Sports Firms"

_sustainability, doi:10.3390/su15086991_

Round 1
Reviewer 1 Report
Title: Article Innovation-driven Policy, Corporate Governance Structure and Total Factor Productivity in Chinese Sports Sector: Evidence from Listed Sports Firms
After reviewing this article, I think it is potential for publication but the authors should revise as comments below:
- In the introduction, the authors should emphasize the motivation and the contribution of the research. Are there any differences from previous studies?
- In the literature review, the authors should review and update recent studies relating to the corporate governance structure. I suggest the authors review and cite some recent studies such as Nguyen (2022a); Dang and Nguyen (2021); Naciti et al (2022); Nguyen (2022b); Nguyen (2020); … (see reference).
- In section 3.1, please explain why the data start at 2015 and stop at 2020
- The use of variables must be explained. On what basis do the authors use those variables?
- In conclusion, the authors should briefly summarize the results of the study, its implications. In addition, the authors need to present research limitations and directions for further research
- There are some typos and grammatical errors, you must check it again carefully.
References
Nguyen, Q. K. (2022a). Audit committee structure, institutional quality, and bank stability: evidence from ASEAN countries. Finance Research Letters, 46, 102369.
Dang, V. C., & Nguyen, Q. K. (2021). Internal corporate governance and stock price crash risk: evidence from Vietnam. Journal of Sustainable Finance & Investment, 1-18.
Naciti, V., Cesaroni, F., & Pulejo, L. (2022). Corporate governance and sustainability: A review of the existing literature. Journal of Management and Governance, 26(1), 55-74.
Nguyen, Q. K. and Dang, V.C (2019). Audit committee structure and bank stability in Vietnam. ACRN Journal of Finance and Risk Perspectives, 8, 240.
Nguyen, Q. K. (2022b). The impact of risk governance structure on bank risk management effectiveness: evidence from ASEAN countries. Heliyon, 8(10), e11192.
Nguyen, Q. K. (2020). Ownership structure and bank risk-taking in ASEAN countries: A quantile regression approach. Cogent Economics & Finance, 8(1), 1809789.
Author Response
Thank you for taking the time to review our manuscript. We truly believe that your constructive comments have helped improve the quality of our manuscript substantially.

Reviewer 2 Report
The similarity ratio is quite high. If the authors are going to reduce this rate, my comment is that the study is well organized overall. Only moderate english changes required.

Author Response
Thank you for taking the time to review our manuscript.
Reviewer 3 Report
Check the analysis and report the results accurately.

Author Response

(The authors gave the same response as above.)

Reviewer 4 Report
Thanks for the efforts by the authors to contribute to the literature. I have reviewed the article carefully and ended up with some serious concerns about the empirical tests.
The authors developed a total seven hypotheses (H1-H7) and tried to test their hypotheses using the regression analysis. But I don't see any other relevant tests including correlations, goodness of fit etc. I raise questions about how did you confirm that the regression analysis is the best to conduct your research? Do you have subsequent tests to justify your methods(regressions) in the study?
Also, from the beginning of the research, your data is not well defined for the study. For example, your sample is 13,303 observations. And you said it's 13,303 firms, but I don't think they are the number of firms. I think it's the nmber of firm-year observations for six years between 2015 and 2020. Then you need to address where or not your sample is balanced panel data in order to confirm your tests.
English: there are many typos and irrelevant terms used throughout the paper. Please take a extensive step for the english editing. For example, the following sentence is not quite understandable.
"Based on the above literature, in the context of China's innovation driven development, supply side structural reform and high-quality development of the sports industry,
this study aims to fill in the gaps by addressing the three questions:"
It is too much redundant.
Minor issues: Please double check the format of the paper. I don't think the authors followed the format of Sustainability. (structure, reference format etc.)
Best,
Author Response

(The authors gave the same response as above.)

Round 2
Reviewer 1 Report
This version is better and can be publised
Author Response

(The authors gave the same response as above.)

Reviewer 3 Report
Editing is required for the references given at the end. The format is inconsistent when it comes to page numbers and improper capitalization.
Author Response
Based on your comments, we have refined the reference section. Thank you for taking the time to review our manuscript. We sincerely believe that your constructive comments help to significantly improve the quality of our manuscripts.
Reviewer 4 Report
I think the authors took care of the revisions in relevant ways.
Great work.
Best,
Author Response
Thank you for taking the time to review our manuscript. We sincerely believe that your constructive comments help to significantly improve the quality of our manuscripts.